# Workplace-Based Interventions for Mental Health in Africa: A Scoping Review

**DOI:** 10.3390/ijerph20105863

**Published:** 2023-05-18

**Authors:** Munira Hoosain, Naafi’ah Mayet-Hoosain, Nicola Ann Plastow

**Affiliations:** Division of Occupational Therapy, Department of Health & Rehabilitation Sciences, Faculty of Medicine & Health Sciences, Stellenbosch University, P.O. Box 241, Cape Town 8000, South Africa

**Keywords:** workplace rehabilitation, mental health, depression, vocational rehabilitation

## Abstract

Mental health problems are one of the leading contributors to the global burden of disease. Workplaces provide a valuable and accessible setting for interventions to improve worker health. However, little is known about mental health interventions on the African continent, particularly those based in the workplace. This review aimed to identify and report on the literature about workplace-based interventions for mental health in Africa. This review followed the JBI and PRISMA ScR guidelines for scoping reviews. We searched 11 databases for qualitative, quantitative and mixed-methods studies. The grey literature was included, and no language or date limits were applied. Title and abstract screening and full text review, were completed independently by two reviewers. A total of 15 514 titles were identified, of which, 26 titles were included. The most common study designs were qualitative studies (n = 7) and pre-experimental, single-group, pre-test, post-test studies (n = 6). Workers with depression, bipolar mood disorder, schizophrenia, intellectual disability, alcohol and substance abuse, stress and burnout were included in the studies. The participants were mostly skilled and professional workers. A wide variety of interventions were offered, of which, most were multi-modal. There is a need to develop multi-modal interventions in partnership with stakeholders, particularly for semi-skilled and unskilled workers.

## 1. Introduction

Mental health problems are the leading cause of absence from work globally, resulting in a multitude of problems for workers, employers and the economy [1]. Mental illness is also one of the largest contributors to the global burden of disease, particularly in adults of working age [2]. In the most recent global burden of disease study, depressive disorders, anxiety and pain were among the ten most important causes of increasing disability-adjusted life years (DALYs). All three increased in the rank of leading causes of DALYs from 1990 to 2019, indicating that mental illness and pain are becoming more common causes of ill health internationally over time. Workplaces can either be harmful to health or promote health [1,3]. Many work environments include a lack of variety of work, fragmented or meaningless work, overload or underload, time pressures, inflexible work schedules, lack of control and decision-making abilities, poor environmental conditions, unsupportive work cultures, poor communication and relationships at work, job insecurity and conflicting demands of home and work. These environments are harmful to health as they increase psychosocial risks [3]. In contrast, since working-age adults often spend the majority of their waking hours at work, the workplace also provides a valuable and underutilised setting for health interventions targeted at promoting and improving worker health [4].

Several systematic and scoping reviews have synthesised evidence about workplace-based interventions for mental health [5,6,7,8,9,10]. However, none could be found that focused specifically on low- to middle-income countries or the African continent. Africa has unique contextual considerations. In the workplace, there is a larger proportion of semi-skilled or unskilled positions than is typically seen in higher-income countries. Gender inequity in the workplace is a particular problem, with women in Africa more likely to work in low-skilled and informal or temporary positions, with worse working conditions than men [11]. In addition, Africa has the smallest proportion of mental health workers internationally [12]. Other challenges to healthcare provision include poverty, mental health stigma and the wide variety of cultures and languages [13]. Limited evidence is available about interventions to support the mental health of workers in the African context. Therefore, this scoping review aims to identify the literature about workplace-based interventions for mental health in Africa. The objectives of the review, as stated in the review protocol [4], are:To provide a detailed overview of all studies about workplace-based interventions supporting mental health in Africa.To identify trends and gaps in the types of interventions, practitioners involved, mental health conditions, types of work, geographic location, anticipated outcomes and effectiveness of interventions.To identify barriers and facilitators in implementing these interventions on the African continent.

## 2. Methods

We used the Preferred Reporting Items for Systematic Reviews and Meta-Analyses guideline for reporting on scoping reviews (PRISMA-ScR) [14] and the updated Joanna Briggs Institute (JBI) guideline for conducting scoping reviews [15] to design and report on this review. The review methods are described in detail elsewhere [4]; a summary is provided here. In accordance with the ACTIVE framework [16], we consulted with various stakeholders to inform the design of this review, including African occupational therapists with an interest in workplace mental health and a group of managers and supervisors at a large South African factory where workplace-based rehabilitation services are provided. 

### 2.1. Search Strategy

The databases searched included Medline (Pubmed), EBSCOhost (Academic Search Premier, AfricaWide Information, SINAHL, Health Source:Nursing/Academic), Scopus, Web of Science, Sabinet, Cochrane and OTSeeker. The grey literature was searched via ProQuest and Sabinet. We applied no date or language restrictions. Reference lists of included articles were searched manually. We also contacted key contributors in the field to personally source the relevant literature.

The search string was developed with the help of a specialist librarian and the Systematic Review Accelerator. A comprehensive account of search strings per database is included in Appendix A, and the search terms are detailed below in Table 1.

### 2.2. Eligibility Criteria

The population, concept and context framework (PCC) was used to define review parameters, in accordance with the JBI guideline [15]. Studies were included if the study population involved people actively engaged in work, preparing for work or returning to work. This included child labour and formal and informal employment. For concept, any research that described interventions that support mental health and was at least partly based in the workplace was included. The review protocol [4] planned to only include interventions offered by occupational therapists, but we soon realised that these were scarce. We therefore included any non-pharmaceutical intervention that met the other eligibility criteria. The search strings were not adjusted, as the “occupational therapy” string included many alternate terms that would capture non-pharmaceutical promotive, preventative and promotive interventions. Studies had to be conducted in an African country, in any type of workplace. We included qualitative, quantitative and mixed-methods studies of primary research. Protocol papers were excluded if no results were discussed. Opinion articles, commentaries and editorials that did not describe or evaluate interventions were also excluded.

### 2.3. Study Selection

Titles and abstracts from the literature search were uploaded to Covidence and screened independently by two reviewers (M.H. and T.C.). Discrepancies were discussed and eligibility criteria clarified after the first 100 titles. Full-text screening was also conducted independently by the same two reviewers. Conflicts were resolved through the discussion and clarification of inclusion and exclusion criteria between M.H., T.C. and N.A.P. The PRISMA flow chart in Figure 1 shows reasons for exclusion at the full-text review stage. Twenty-four articles from the initial search were included in the review, and two additional titles were identified through pearling.

### 2.4. Data Analysis

Data were extracted manually by two researchers (M.H. and N.M.) onto Excel spreadsheets, and these data were mapped by geographic location, study design, types of workers and intervention particulars, in accordance with the study objectives. Barriers and facilitators to intervention were identified across studies, along with trends and gaps in the existing research. In keeping with the scoping review methodology, we did not analyse the methodological quality of the included articles. A meta-analysis was not completed, as this is not normally carried out with scoping reviews. This was also not possible due to the heterogeneity of the data.

## 3. Results

As can be seen in Figure 1, the literature search yielded 15,514 titles, of which, 4106 were duplicates. We thus screened 11,408 titles and abstracts, of which, 11,305 were irrelevant. In total, 103 full-text studies were reviewed, and 24 of these were included in the scoping review. Two additional titles were identified for inclusion from the reference lists of included studies, making twenty-six titles for inclusion in this review. 

### 3.1. Characteristics of Included Titles

Of the 26 titles included in this review, 18 were peer-reviewed journal articles, 6 were dissertations, 1 was a letter published in the *South African Medical Journal* and 1 was a magazine article published in *HR Future*. As can be seen in Table 2, the majority of studies and articles related to South Africa (n = 23). Other countries represented were Botswana (n = 2) [17,18] and Kenya (n = 1) [19]. Publication dates varied from 1995 to 2021, and there were a variety of African and other international journals.

### 3.2. Type of Studies

A variety of study designs were represented, including randomised controlled trials (n = 3) [21,22,23], quasi-experimental studies with two groups and no randomisation (n = 2) [27,39] and pre-experimental single group studies with a pre-test, post-test design (n = 6) [24,26,27,31,34,38]. There were also several qualitative studies (n = 7) [24,34,35,44] and mixed-methods designs (n = 4) [24,36,41], as well as cross-sectional quantitative studies (n = 2) [18,20] and a longitudinal descriptive study (n = 1) [42]. 

### 3.3. Population Characteristics

The types of workplaces represented in the included titles varied. The majority of titles related to hospitals and clinics (n = 8) [17,20,25,28,36,37,40,42,44], as well as corporate environments (n = 8) [19,23,32,39,40,41,42]. These were followed by manufacturing and industrial workplaces (n = 4) [22,23,24,29,35], schools and universities (n = 4) [25,26,38,41] and a municipality (n = 1) [28]. The remaining two studies were conducted in various workplaces [32,43]. 

No articles on child labour were found. All titles related to adults of working age. Most titles related to professional, skilled or semi-skilled workers. Many studies included workers with no health conditions. Mental health conditions represented in studies’ inclusion criteria were schizophrenia, depression, bipolar mood disorder, intellectual disability, burnout, alcohol or substance abuse and high levels of stress.

### 3.4. Interventions and Outcomes

A large variety of treatment modalities were discussed in the included titles. These were offered by a range of practitioners, including psychologists, occupational health nurses, occupational therapists, physiotherapists, social workers, doctors, counsellors and other healthcare workers. Both group and individual interventions were offered. Treatment modalities included relaxation and stress management techniques, cognitive behavioural therapy (CBT), transpersonal psychology, rational-emotive therapy, psychoeducation, counselling, health promotion and workplace wellness initiatives, physical exercise and fitness programmes, biofeedback devices, life skills training, energy conservation techniques, roleplay, environmental adaptations such as “green environments” and work rehabilitation. Work rehabilitation initiatives included protective and supported employment, energy conservation techniques, work simplification, work retraining, work preparation and return-to-work programmes. Alternative therapies such as Indian head massage, yoga, Reiki and crystal healing were discussed in two titles. One study made reference to the promotion of leisure-time activities as a treatment modality [31].

Theoretical frameworks for interventions included the Model of Human Occupation, Social Exchange Theory, Work Integration Social Enterprise framework, Gestalt theory, Supported Employment model, the Clubhouse model, Cognitive Behavioural Therapy, Paulo Freire’s adult education theory and Transpersonal psychology. 

Not all titles discussed outcomes. Some of the outcomes studied in the included titles were stress [18,21,24,31,34], mood [24], quality of life [24], absenteeism and presenteeism [23,39], exercise behaviours [21,23,27], coping skills [24,34], job satisfaction [18,39] and participation in activity [18,39].

### 3.5. Effectiveness of Interventions

Common outcomes of the interventions across the included titles were self-reports of increased motivation, improved interpersonal relationships, reduced stress and burnout, higher optimism and improved efficacy. One particular title reported that their workers were able to regulate their own emotions and trauma responses, which then enabled them to provide improved and regulated support to their colleagues and clients [45]. Among the included interventions, the modalities which were found to be effective were psychoeducation, relaxation, play, health promotion and lifestyle management. It is important to note that titles with high effectiveness tended to utilise a multi-modal approach to treatment. 

One title reported that employees did not experience positive outcomes due to a lack of organised and efficient interventions [33]. The limited involvement of stakeholders and employees in the planning and implementation process of the intervention may render the intervention less specific and considerably less effective. Additionally, when employers are not involved and invested in the intervention, the employees may find it difficult to prioritise the intervention over their increasing workload. Some titles reported that while the intervention was effective, it did not take priority over the increasingly high workload the employees had experienced [18,21,25].

Two titles assessed outcomes over multiple time points to determine long-term effectiveness [21,22], while the majority of the titles assessed outcomes at baseline and one additional time point. 

### 3.6. Barriers and Facilitators

A number of barriers to workplace-based interventions for mental health were identified. Stigma was a barrier to participant recruitment and participation in programmes [22,37,46]. Employers were not always on board with the programmes, and interventions were sometimes poorly implemented as a result [22,33,37,46]. Limited long-term follow ups of outcomes after the programmes was another barrier [34,37,39,41,43]. Reported barriers to research on workplace-based interventions for mental health included small sample sizes, study design and limited generalisability [21,22,24,26,27,28,31,32,34,35,36,38,39,41,42,43]. Authors also named outcome measure limitations as a barrier. Appropriate outcome measures were not always available, and these were often self-report measures [23,26,27,29,31,33,35,37,41,43,47,48]. Time constraints was another barrier, both to the interventions and to the research [26,37,39].

Workplace-based interventions were successful when integrated into the organisation’s culture, when these were designed by diverse committees, and when administrative support was available [3,5,17,36,40]. Other facilitators included a strong theoretical framework for interventions, multi-modal interventions, short duration, cost-effectiveness and feasibility [24,26,31,34,37,41,42,47]. Accessibility to participants was another facilitator, as interventions were all based in the workplace [17,21]. Research facilitators included multiple research designs in mixed-methods studies, large sample sizes and previously validated outcome measures [10,20,26,33,38,46].

### 3.7. Service Users’ Perspectives

Service user experiences and perspectives were overwhelmingly positive across the studies. Participants reported reaping benefits and enjoyment from the interventions [17,24,25,26,27,31,32,38,42,43,44]. Several studies reported that participants were motivated to engage in and continue with interventions, and they were satisfied with outcomes [23,35,36,45]. They reported gaining skills and finding meaning within interventions [17,24,25,26,27,31,32,38,42,43,44]. Some studies reported that participants did not engage optimally in the programme due to high workloads [17,21]. Some participants were nervous about the stigma that could be associated with the programme, as it was associated with a mental health condition or practice area [22,37,46]. Only one study reported negative participant experiences, in which participants felt that the programme was poorly run and implemented [33].

## 4. Discussion

This scoping review aimed to identify and map the existing literature about workplace-based interventions for mental health in Africa. While several systematic and scoping reviews have been conducted in the field of mental health interventions in the workplace [5,6,7,8,9,10,45], this is the first to highlight the African literature and interventions tailored to unique considerations of the African context.

The majority of the 26 included studies were conducted in South Africa, indicating an under-representation of other African countries. Intervention studies tended to have mostly single-group, pre-test, post-test pre-experimental study designs. There was also a large proportion of qualitative studies. The populations included in the studies were mostly professional and skilled workers, with a minority of unskilled workers. There was a wide variety of treatment modalities, including many multi-modal interventions, offered by various different service providers. Salient barriers and facilitators were buy-ins from employers and managers and accessibility to participants in the workplace. Service user perspectives and experiences of these interventions tended to be mostly positive, with participants experiencing meaning and enjoyment and gaining skills during interventions. 

The World Health Organization’s recently published guidelines on mental health at work [3] recommend interventions at various levels, including organisational interventions, training managers in the workplace, training workers in mental health literacy and awareness, individual interventions, returning to work after mental illness, gaining employment with mental illness and screening programmes. This review found that African interventions in the workplace are situated at several of these levels, as interventions were found to be within organisations and across organisations, individual and group-based. The study by Soeker et al. [35] described the promotion of leisure-time activity as a mental health intervention, which is also supported by the WHO guidelines [3]. The prominence of mental health interventions in health institutions is also in line with the WHO guidelines, as health, humanitarian and emergency workers have been identified as an at-risk group with particular need for psychosocial interventions [3]. It is interesting to note that there were very few titles relating to semi-skilled and unskilled workers, such as factory workers. This group of workers may currently be under-represented in mental health interventions, possibly due to stigma related to participation in mental health interventions. 

The wide variety of treatment modalities and multi-modal interventions is supported by other reviews, which also found that multi-component interventions had greater evidence to support their effectiveness [8,10]. As Cullen et al. reported, psychosocial interventions such as cognitive behavioural therapy on their own, without the inclusion of supplementary treatment modalities such as workplace modifications or accommodations, do not seem to be effective [10]. Practitioners and researchers who are developing interventions should thus consider including multi-modal interventions to address the complex nature of mental health at work. Future studies should monitor outcomes over time to reflect the long-term effectiveness and feasibility of workplace interventions. The importance of buy-ins from managers and supervisors as a potential barrier or facilitator is not surprising. The importance of stakeholder engagement in intervention development has been well established. Collaborative interventions tend to have better outcomes, particularly where stakeholders have been included from the beginning [45,48,49,50,51]. Engagement with service users and gatekeepers such as managers, supervisors and occupational health staff in workplaces in the intervention design and implementation allows for interventions to be culturally appropriate, contextually relevant, feasible and acceptable. It is likely that perspectives and experiences of service users will be more positive with greater engagement in the initial phases of intervention design. 

### Strengths and Limitations

This scoping review was strengthened by the independent duplicate screening and full-text review of articles for inclusion. All languages and dates were included, and the grey literature was searched extensively. While the literature search was broad and extensive, using many databases and alternate search terms, it is possible that relevant titles may have been missed. Non-electronic databases were not included, and therefore, dissertations and theses held at institutions that were not published were not sourced. The quality assessment of individual studies was not completed, as is typical with systematic scoping reviews.

## 5. Conclusions

A broad range of workplace-based treatment modalities to promote and improve mental health are being provided in South Africa, suggesting that workplaces are a promising environment for mental health promotion. However, limited research was found outside of South Africa, highlighting a need for interventions and research in other areas of the continent. Workplace-based interventions for mental health should be multi-modal, considering psychosocial interventions as well as work-related interventions such as workplace modifications or accommodations. This review further highlights a critical need for more implementation research in Africa. Future research should investigate the effectiveness and feasibility of interventions, as well as the acceptability, accessibility and uptake of interventions. The geographical reach of future studies should be expanded outside of South Africa. Future studies should also be aimed at unskilled and semi-skilled workers, since most studies included skilled workers. Service users and managers in workplaces should be involved in intervention design and implementation as much as possible to increase the uptake and acceptability of interventions. Outcomes should be assessed at more than one time point post-intervention to monitor long-term effectiveness and feasibility. It is further recommended that the Template for Intervention Description and Replication (TIDieR) checklist and guide are used to describe interventions, along with the Standard Protocol Items: Recommendations for Intervention Trials (SPIRIT) statement [52,53]. The findings of this review suggest that the burden of disease caused by mental health problems in Africa may be effectively addressed by providing multi-modal workplace-based interventions that are developed in collaboration with stakeholders.

## Figures and Tables

**Figure 1 ijerph-20-05863-f001:**
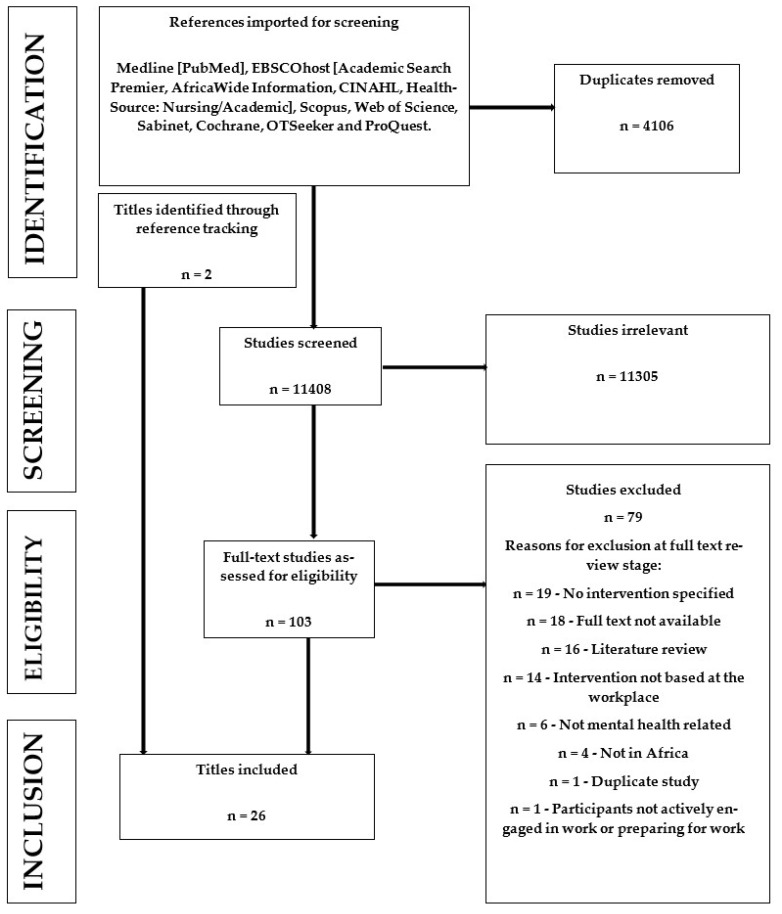
PRISMA Flow Chart.

**Table 1 ijerph-20-05863-t001:** Search strategy.

Concept	Alternative Words
Mental health	“Mental disorder*” OR burnout OR stress OR psychosocial OR wellbeing OR well-being OR wellness OR recovery OR “substance abuse” OR “alcohol abuse” OR “drug abuse” OR “post traumatic stress disorder” OR “post-traumatic stress disorder” OR PTSD OR depression OR anxiety OR schizophrenia OR suicide
Occupational therapy	“occupational therap*” OR rehabilitation OR prevention OR promotion OR habilitation OR assessment OR “supported employment” OR “return-to-work” or “return to work” OR treatment OR intervention OR effectiveness OR effect OR counselling OR “work ability” OR mindfulness OR “nature-based”
Workplace	“workplace based” OR “workplace-based” OR organisational OR organizational
Africa	Africa OR Botswana OR Ghana OR Kenya OR Madagascar OR Malawi OR Mauritius OR Morocco OR Namibia OR Nigeria OR Rwanda OR Seychelles OR “South Africa” OR Tanzania OR Tunisia OR Uganda OR Zambia OR Zimbabwe

**Table 2 ijerph-20-05863-t002:** Characteristics of Included Titles.

Author and Year, Journal	Country	Study Design (Sample Size)	Intervention Treatment Modalities	Type of Workplace
**Basson et al., 2013,** [20] *British Journal of Nursing*	South Africa	Quantitative Descriptive (n = 141)	Stress management techniques; reconstructed working hours to allow for rest; protection initiatives, counselling; education on health hazards; lifestyle changes; creating a supportive work environment	Hospital
**Broome 1995,** [21] Dissertation	South Africa	RCT (n = 374)	Relaxation techniques (Transcendental Meditation and Progressive Muscle Relaxation); education; CBT	Marketing research consultancy
**Burnhams et al., 2015,** [22] *Substance Abuse Treatment, Prevention and Policy*	South Africa	Cluster RCT (n = 325)	CBT; group intervention; health promotion	Safety and security employees in a municipality
**Edries et al., 2013,** [23] *BMC Public Health*	South Africa	RCT (n = 80)	Health promotion; physical exercise; pamphlets; goal-setting	Clothing factories
**James 2003,** [24] Dissertation	South Africa	Pre-experimental: single group, pre-test, post-test study (n = 14)	Psychoeducation; physical fitness; CBT	Coal terminal (industrial site)
**Johnson et al., 2013,** [25] *South African Journal of Psychology*	South Africa	Pre-experimental: single group, pre-test, post-test study (n = 30)	Transpersonal psychology techniques	School
**Johnson et al., 2017,** [26] *AIDS Care*	South Africa	Qualitative (n = 27)	Transpersonal psychology techniques; burnout program; psychoeducation; skills training	School
**Kennedy et al., 2008,** [27] *Journal of Workplace Behavioral Health*	South Africa	Pre-experimental and quasi-experimental (n = 39)	Biofeedback devices relaxation and deep breathing; psychoeducation	Call centres
**Ledikwe et al., 2017,** [17] *Journal of Environmental Medicine*	Botswana	Mixed methods (n = 38)	Workplace wellness activities	Healthcare facilities
**Ledikwe et al., 2018,** [18] *BMJ Open*	Botswana	Quantitative cross-sectional (n = 1318)	Health screening, treatment and care; health promotion; stress management and team building; occupational health and safety; psychosocial and spiritual care and therapeutic recreation	Hospitals and clinics
**MacDougall et al.,** [28] **2021,** *International Journal of Mental Health*	Kenya	Qualitative (n = 7)	Psychoeducation; psycho-social skills training; supported housing; education, employment and leisure; community-based rehabilitation; work integration	Unemployed, volunteers or sporadic employment
**Mchunu et al., 2008,** [29] *Occupational Health Southern Africa*	South Africa	Exploratory case study, mixed methods (n = 258)	Psychoeducation; health promotion; health management in lifestyle; adaptive environment changes	Small, medium and large organisations (private manufacturing, production, engineering; parastatal engineering and academic and private health)
**Mendonca 2003,** [30] *HR Future* (magazine)	South Africa	Magazine Article	Indian head massage	Corporate offices
**Nel 2006,** [31] Dissertation	South Africa	Pre-experimental: Single group pre-test post-test design (n = 12)	Play; relaxation; sensory play	IT department in bank
**Ordman 2001,** [32] Dissertation	South Africa	Qualitative Content Analysis (n = 11)	Medication (anti-depressants, tranquilisers, anti-psychotics and other); back-to-work rehabilitation; protective employment	Sheltered workshop with psychiatric patients
**Rakepa et al., 2013,** [33] *African Journal of Public Affairs*	South Africa	Mixed Methods (n = 300)	Psychoeducation; counselling; advisory services; CBT; screenings	Corporate environment
**Reingold 1999,** [34] Dissertation	South Africa	Pre-experimental: single group pre-test post-test design (n = 24)	Rational-emotive therapy, accelerated personal development program	Emergency assistance company
**Soeker et al., 2016,** [35] *Work*	South Africa	Qualitative descriptive (n = 3)	Work simplification; energy conservation techniques and ergonomic analysis techniques; work retraining; group therapy; leisure-time activity promotion; financial management education	General assistants in soft drink manufacturing; perfume company; retail store
**Soeker et al., 2021,** [36] *Work*	South Africa	Qualitative descriptive (n = 5)	Return to work; group therapy	Psychiatric hospital
**Steenkamp 2008,** [37] Dissertation	South Africa	Mixed Methods (n = 325)	CBT; psychoeducation; adult learning and teaching; roleplay	Public hospital
**Taute et al., 2010,** [38] *Social Work*	South Africa	Pre-Experimental: Single group pre-test post-test design (n = 71)	Group therapy; psychoeducation; CBT; skills training; behaviour modification	Tertiary education institution
**Thatcher et al., 2012,** [39] *Work*	South Africa	Quasi-experimental (n = 240)	Environmental adaptations (“green” building)	Financial institution
**Thom 2020 et al.,** [40] *South African Medical Journal*	South Africa	Letter to SAMJ	CBT; education; counselling; health promotion	Healthcare facilities
**Toerien et al., 2020,** [41] *Health SA Gesondheid*	South Africa	Qualitative descriptive (n = 5)	Psychoeducation; group therapy; CBT	University
**Van Niekerk et al., 2015,** [42] *Work*	South Africa	Longitudinal Descriptive (n = 29)	Work training; CBT; group therapy	Psychiatric ward, protective workshop
**Vyas-Doorgapersad et al., 2015,** [43] *Gender & Behaviour*	South Africa	Qualitative (n = 31)	Alternative therapies: yoga, meditation, crystal healing and Reiki	Various, (yoga instructor, self-employed, bank employee, educator, fitness trainer)

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
