# Peer review of "Workplace-Based Interventions for Mental Health in Africa: A Scoping Review"

_ijerph, 2023, doi:10.3390/ijerph20105863_

Round 1
Reviewer 1 Report
This is a very interesting scoping review, addressing a strong theme of workplace well-being in the African context.
The authors discuss the theme satisfactorily under the scientific quality, adopting a rigorous methodology such as PRISMA.
So considering I agree to accept the paper in its current form.
Author Response
Dear Reviewer 1
Thank you very much for your kind comments below. We see that you have recommended no revisions, and thus have not attached a revised manuscript in this response.
Kind regards
The authors
Reviewer 2 Report
First of all, congratulations to the authors to pick a topic that is both underexplored, and very much worth exploring. Too often the existence of a 'gap' doesn't mean a desperate need to fill it. Looking at a developing world context, where the pressures to have a job are more life-and-death than in the developed world where various safety nets exist, is genuinely valuable. I also congratulate the authors on a cogent and compact introduction, and a clear well put-together method to their review. I note that the research methods are described elsewhere, but I do think we need to see what key words you used to reach the results--some of the other details can be ommitted. To a significant degree, the question you 'ask' a database determines the answers, and we don't know what question you asked, unless we read an ancilliary paper. I had a look at Table 1 of the earlier paper--I believe that can be included here. I thought it was remarkable that so many papers were caught in the review 'net' at the first cast, and then only 26 were actually about interventions. I think that says something about the state of the scholarly literature...
Overall, while the paper found relatively little literature, and the literature that was found is quite skewed and even collectively does not add much to our understanding of interventions in Africa, this paper has been nicely executed. I think the only changes I can suggest are as follows:
a] In the conclusion it states that the study suggests "that workplaces are a promising environment for mental health promotion" and "a broad range of workplace-based interventions" are being provided in Africa--but neither statement is supported by the literature review. We have no idea of the prevalence of such interventions, considering that the studies were almost all South African, and almost all devoted to white collar jobs...and in that sense suggests the literature review's findings should not be all that different to one conducted, say, in Spain or Greece..
b] Similarly, the recommendations for future research should surely include expanding the geographical reach of the studies, and a shift to include interventions (if any) in blue collar jobs.
c] The inclusion of the keywords table as noted earlier.
d] I felt the analysis of the papers was hampered by a sense (which may be misplaced) that a deeper thematic sifting/synthesis of the papers had not been undertaken. Certainly, in reporting your results you frequently use terms like "several studies" and "authors also named outcome measure limitations" and so on...without letting us know WHICH papers had these features. With the numbered referencing style, it should be possible to include this information quite compactly.
Overall an interesting project worth publication.
Reviewer 3 Report
Thank you for your work with this unique scoping review. Overall, the review is an interesting study that the readers of IJERPH. I have offered a major suggestion to further strengthen the paper, for your review and consideration again.
Title
The title is not precise and valid meaning. A precise could be "Workplace-Based Interventions to Mental Health Problems: A Scoping Review"
Abstract
The abstract is not clearly identified the problem statement of your aims, a brief description of the scoping review, the key findings, interpretation of the key results, and some conclusion/implications.
My question is, what your aims are? What methods are? What the findings are? What originality is? Should be clearly defined these issues. See the abstract section.
Introduction
Clarify what the introduction through objectives are. Should be clear clarified what gap of study you try to review? What dimensions behind the mental health problems in Africa are? What dimensions are you reviewing?
See line 26-63
General problem statement problems are not identified. The introduction is too limited and very short. Should provide a clearance issue related to gap of phenomena/theory/concept/issue provide to review. The objectives are too limited. Should be added more what/how you provide to study.
Theoretical lens
Should be provided more information about the definition of mental health, workplace-based interventions, and Africa contexts.
Methods
The methods are too limited in scoping review approach. The design is too general (line 65-72). The searching strategy is not specific key terms (line 75-83). What methods of scoping review approach are? How did you analyse the secondary data?
Methods
The interpretation of the results is not logical order data.
3.1. Characteristics of included titles are too limited. Should be more explained what really characteristics are?
3.2. Type of studies are too short in explaining. Should be provided about what types relate to scope review?
3.4. Interventions and outcomes. What and how scoping reviews approach to interventions and outcomes. I found the interventions are to general information. Should be cleared what and how?
Discussion
Discussion is too short in debate and discussing with the main study and others scholars. See line 214-260.
Could be added more information about the (1) theoretical scoping contributions, (2) practical implications, (3) mental health practices.
Round 2
Reviewer 3 Report
All comments are revised and well-suited for publication. I accepted as a revised version format. Good luck!